# Evaluation of Land Use Efficiency in Tehran's Expansion between 1986 and 2021: Developing an Assessment Framework Using DEMATEL and Interpretive Structural Modeling Methods

Safiyeh Tayebi [1,*,†], Seyed Ali Alavi [2,†], Saeed Esfandi [3], Leyla Meshkani [4] and Aliakbar Shamsipour [1]

1    Faculty of Geography, University of Tehran, Tehran 1417853933, Iran
2    Faculty of Sciences and Bioengineering Sciences, Katholieke Universiteit Leuven, 3000 Leuven, Belgium
3    Center for Energy and Environmental Policy, Joseph R. Biden, Jr. School of Public Policy and Administration, University of Delaware, Newark, DE 19716, USA
4    Faculty of Earth science, Shahid Beheshti University, Tehran 1983969411, Iran
*    Correspondence: tayebi.s@alumni.ut.ac.ir
†    These authors contributed equally to this work.

**Abstract:** This paper aims to reveal the shortcomings of the land use efficiency assessment formula presented in SDG 11.3.1 Indicator and develop a framework that can provide urban planners with a more accurate understanding of the variables influencing and/or influenced by urban expansion. Based on the mentioned formula, Tehran never experienced urban shrinkage between 1986 and 2021, as shown by the relationship between land consumption and population growth. However, the research findings indicate that land allocation patterns have not only decreased most urban services per capita, but have also undermined ecosystem services during this period. In this paper, we propose a new assessment framework by which a dual aspect of urban planning is addressed, namely providing sustainable urban services while protecting natural resources, and using ecosystem services sustainably to support cost–beneficial urbanization. For this purpose, a total of ten mainly repeated contributing variables were collected in the categories of environmental, physical-spatial, and economic–social effects of urban expansion. A questionnaire based on these variables was prepared, and 14 urban planning experts collaborated to classify the variables and identify causal relationships between them. In the following, data obtained from the questionnaires were analyzed using DEMATEL and Interpretive Structural Modeling (ISM) methods to determine which variables influence and/or are influenced by urban expansion (and to what extent). Third-level variables that directly influence urban expansion include transportation (A6), infill development (A7), and entrepreneurship (A10). Spatial justice (A8) and housing and population attraction (A9) were identified as middle-level variables that both affect and are affected by urban expansion. Finally, land surface temperature (A1), air pollution (A2), sewage and waste (A3), water resources (A4), and vegetation (A5) were identified as first-level variables that are mainly affected by urban expansion.

**Keywords:** urban expansion; land use efficiency; SDG indicator 11.3.1; ecosystem services; urban services; Tehran

## 1. Introduction

Based on the United Nations demographic report [1], over the next half-century, the rate of urbanization in the world is projected to reach 68.4%, increasing from 56.2% in 2020 [2]. Although this urban population growth has been affected by socioeconomic and environmental factors in different parts of the world, its impacts on urban expansion patterns have not been sufficiently addressed [3]. Because of their mutual relationship with providing urban services per capita for accessibility, citizen welfare and livability on the one hand and consuming natural resources and using ecosystem services on the

other, these patterns are critical in arranging population attraction, territory planning, and urban management [4–6]. Furthermore, urban expansion has always been considered one of the most critical challenges in land use planning, since urban sprawl and low-density development, as well as geographical divisions of essential land uses, have all contributed to an increase in public and private costs [7]. That is why urban management plays a vital role in determining frameworks for population attraction and distribution patterns in urban areas, as well as planning land use and determining urban services per capita and accessibility in order to maximize efficiency at the lowest cost. To address these theoretical and practical challenges associated with land use planning around the world, as well as the need to resolve existing conflicts in order to provide urban services and citizen welfare while preserving natural resources and providing livable societies, the new urban agenda was developed to provide more appropriate policies to achieve sustainable development goals [8]. Among these goals, SDG 11 emphasizes the sustainable development of urban areas. This goal aims to address the challenges of providing affordable housing, access to sustainable transportation, access to open and green spaces, spatial justice, and alleviating urban poverty and land use inefficiency. Therefore, it is anticipated that substantial financial trends are likely to mobilize towards investments in housing, infrastructure, and economic development to support the achievement of the eleventh sustainable development goal [9]. In response to this issue, urban centers may attract a larger population and expand rapidly, which is likely to increase urban congestion diseconomies [10,11]. Ultimately, the vital issue is aligning these policies in a way that maintains and improves the harmony between the provision of urban services and the conservation of natural resources, avoiding environmental damage and eliminating contradictions in order to achieve sustainable development. The realization of this goal requires strategic planning for sustainable cities and the assessment of progress toward sustainable urban development [12].

Prior to the new urban agenda and sustainable development goals, there were various theoretical and numerical models for urban land use efficiency assessment. For instance, Steinger et al. [13] and Merino-Saum et al. [14] apply local urban sustainability as a measure of land use efficiency, while Koroso et al. [15] consider urban density as the determiner of the level of efficiency. Researchers also use the amount of urban border expansion [16] and the proportion of barren land [17] in their evaluations. Generally, analyzing land use efficiency primarily relies on input–output balances and correlation models. Land use efficiency assessment studies can be divided into four categories: calculations, location and time characteristics, contributing variables, and improvement methods [18] as the most holistic ones. The United Nations developed the 11.3.1 indicator of land consumption rate to the population growth rate (Table 1). This indicator refers to the amount of land consumed by a city to accommodate its growing population and respond to this population's growing needs. In an ideal situation, both variables should be synchronized and measured in a time frame that can be compared. This indicator was designed to measure Urban Land Use Efficiency for sustainable urban development and demonstrate whether urban areas are being used to meet social–economic and environmental needs [19]. In fact, the issue is investigating the ratio of the multi-dimensional and actual demand for land and land dedication [20].

**Table 1.** SDG 11, Target 11.3, and Indicator 11.3.1 [21].

| 11. Goal | 11.3. Target | 11.3.1. Indicator |
|---|---|---|
| Make cities and human settlements inclusive, safe, resilient and sustainable | By 2030, enhance inclusive and sustainable urbanization and capacity for participatory, integrated and sustainable human settlement planning and management in all countries | Ratio of land consumption rate to population growth rate |

This indicator has been criticized, and some improvements have been proposed. In spite of the simple criteria and methodology used to calculate the UN sustainable

development indicator 11.3.1, critics claim that obtaining this information on a global scale is challenging. They also criticize the fact that this indicator only considers built-up area and population and disregards other influencing factors [9,19] such as density variation in different urban regions. Additionally, there is considerable uncertainty regarding the concept of built-up areas in different land use classification systems. A fundamental difference lies in the definition of public space (including parks, gardens, and roads), which affects assessments of built-up areas [19]. A number of contexts have considered the fact that SDGs are not local and that governments and financial levels have different interpretations of implementing SDGs and reporting on them [22–24]. Furthermore, it can be argued that if the final aim of all the indicators of the 17 goals is to ensure sustainable development [22], indicator 11.3.1 is not accurate enough to assess the sustainability of urban expansion and land use patterns due to neglecting urban expansion effects, reduction in ecosystem ability and ultimately neglecting the carrying capacity of the environment [9,25–28]. In other words, the environmental carrying capacity for population growth and urban expansion has been neglected even in its most proportionate condition since this indicator does not consider any limitations for urban expansion and population growth.

Urbanization results in irreversible changes in urban landscapes and adversely influences biodiversity due to its large ecological footprint and destruction of natural land cover and vegetation [9]. These effects may vary depending on where they occur or how intense they are. Furthermore, not all social classes are equally affected by civilization's effects on ecosystem services. By identifying those regions at risk from urban growth, recognizing the different social demands for ecosystem services among social–demographic groups, and understanding how cities affect ecosystem services, various viewpoints can be developed in this field based on different location contexts [22]. In a systematic literature review intended to identify the most common sustainability assessment goals and frameworks, Cohen [12] concluded that urban sustainability assessment generally does not have an integrated framework. In other words, this paper shows that local characteristics can affect sustainable development's conceptual and calculation frameworks.

Accordingly, this study takes into account the accuracy and localization of indicator 11.3.1 in order to optimize the assessment of land use and consumption efficiency in Tehran. An accumulation of theoretical literature and the expertise of specialists in urban-related fields was used to develop a framework for assessing land use efficiency that can be generalized based on the dichotomy of service provision and preservation of sources and allowing for context differences in land use assessment. Providing a framework like this can help accomplish the third target of the eleventh sustainability goal by adapting it to the conditions and procedures of urban management in developing countries.

## 2. Study Area

Iran's capital, Tehran, with an area of 750 square kilometers and a population of 9 million (according to the 2016 national census), is the world's thirty-eighth most crowded city [29]. Tehran has 22 municipal districts and 353 neighborhoods in which more than 3 million households live. The average population density in Tehran is 205 people per hectare, and the southern half of the city has the highest population density (Figure 1). Natural growth is not the main reason for Tehran's population growth, but rather immigration. According to the data of the latest Iran census, which dates back to 2016, Tehran province is at the top of Iran's immigrant-receiving provinces with a share of 20.2 percent of immigrants. The underlying reason behind this immigration rate is that there has never been room for spatial justice and equitable distribution of resources and activities in national, regional and local plans and programs. By failing to consider the potential of mid- and small-sized cities and rural areas, this issue has not only led to the rapid growth of Tehran and a few other major cities, but has also caused significant damage to the rest of the country. This trend has also been boosted by the high potential for business activities and job opportunities in this metropolis compared to other parts of the country. Hamdi and Fathi [30] examined the causes of migration to Tehran based on economic, social, cultural,

and political factors. Based on the results, Tehran's economic factors are the most influential factors in migration. Next is the economic condition of origin, followed by the social status of Tehran. Moreover, Shahnazi and Etezar [31] studied indicators of income, education, housing, employment, and specialized professions in the cities of Iran and found Tehran to be the most prosperous province economically and socially.

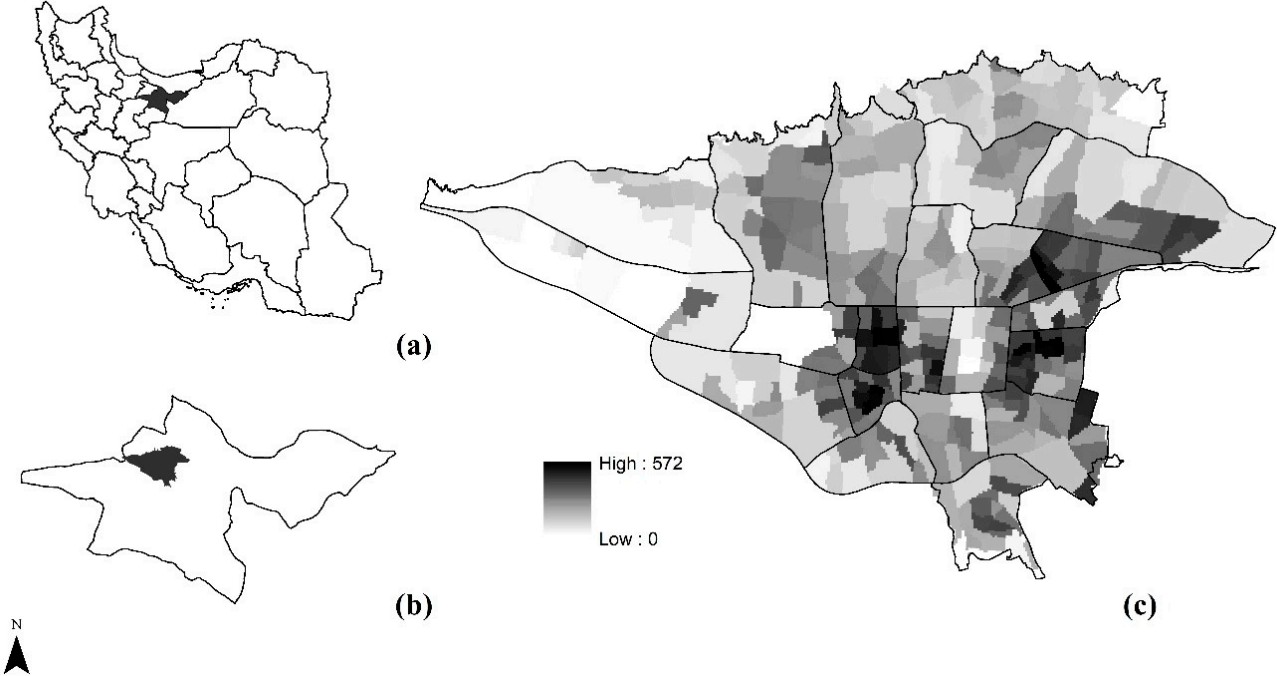

**Figure 1.** The study area: (**a**) Iran; (**b**) Tehran Province; (**c**) Tehran City and its neighborhoods' population density.

　　Various housing policies, concentration of commercial and industrial activities, and lack of holistic and integrated national, regional and local territory planning led to a huge number of immigrants coming to this city. Therefore, Tehran's land cover and land use have undergone considerable changes to meet the needs of the ever-growing population. One of the most noticeable changes is the horizontal expansion of the city due to population growth and increased demand for housing in the past five decades. Figure 2 shows how the city of Tehran expanded from 1921 to 2021 in a 100-year period. Urban growth has also transcended urban boundaries as shown in Figure 2. Nevertheless, the calculations in this paper are based on the area and population of Tehran within its legal (approved) boundary.

　　In the last three decades, city expansion has taken the form of transforming informal and marginal areas into formal towns or new municipal districts. With the justification that more land is required to settle more people within the city, Districts 21 and 22 were officially added to Tehran municipality districts in 1992, marking one of the most significant efforts to expand Tehran's boundaries. This attachment was also performed with the aim of alleviating Tehran's lack of urban services per capita. The residents of these two districts have, however, been struggling with numerous problems due to inadequate infrastructure and services [32]. Table 2 shows the average services per capita in Tehran and the number of neighborhoods with a lack of services. In Tehran's Master Plan, the accessibility of urban services, as a major criterion of quality of life, is evaluated based on the number of these eight urban services per capita. Municipalities are always trying to improve these services in urban neighborhoods. The changes in urban services per capita over time do not show a meaningful trend regarding improved access of citizens to services, as if the blatant growth of the city and the consumption of urban land have not been enough to meet city residents' needs. Tehran also has 4426 hectares of deteriorated areas, representing 7% of its total area, where 20% of the city's population lives. A total of 85% of Tehran's deteriorated areas are

located in the southern part of Enghelab Street, an arterial street that is used as a benchmark that separates the north and south of Tehran. In deteriorated areas of Tehran, population density is three times greater than the average and urban services per capita are one fourth of those in the northern half [33]. Considering all of the above, a large part of the city is lacking services, and population policies and planning services need to be improved.

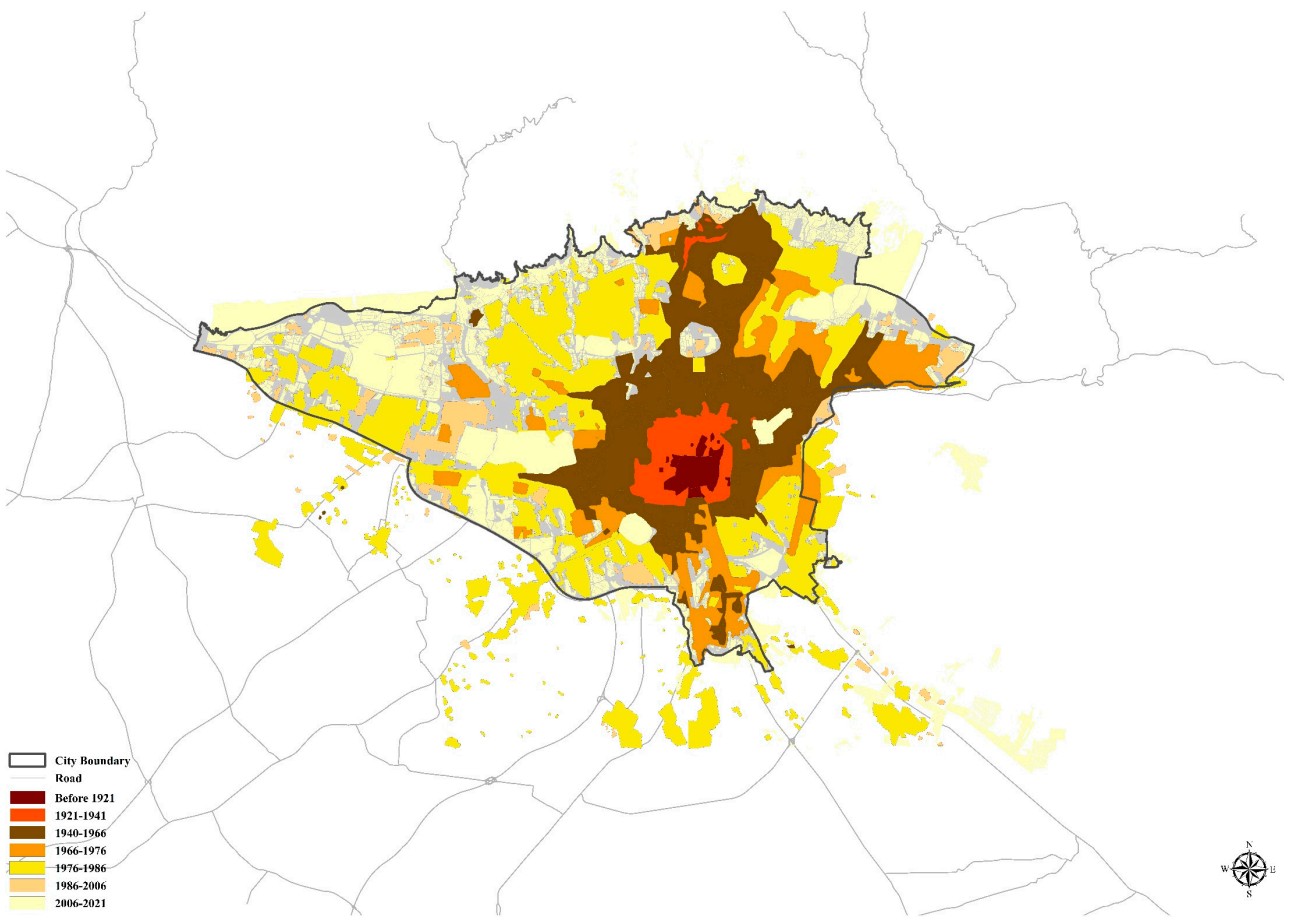

**Figure 2.** Tehran's expansion pattern.

**Table 2.** Tehran's eight main urban services per capita in the 2006, 2011, and 2016 censuses.

| Year | Population | Education Services Per Capita | | Healthcare Services Per Capita | | Urban Equipment Per Capita | | Recreational Services Per Capita | | Cultural Services Per Capita | | Religious Services Per Capita | | Sports Services Per Capita | | Green Spaces Per Capita | |
|---|---|---|---|---|---|---|---|---|---|---|---|---|---|---|---|---|---|
| | | Ave. | N | Ave. | N | Ave. | N | Ave. | N | Ave. | N | Ave. | N | Ave. | N | Ave. | N |
| 2006 | 6,058,207 | 1.58 | 114 | 0.75 | 251 | 1.17 | 252 | 1.63 | 298 | 0.49 | 313 | 0.13 | 243 | 0.13 | 336 | 11.84 | 176 |
| 2011 | 7,803,883 | 1.71 | 114 | 0.76 | 256 | 1.29 | 251 | 2.33 | 298 | 0.49 | 316 | 0.21 | 240 | 0.14 | 336 | 9.89 | 172 |
| 2016 | 9,052,868 | 1.49 | 125 | 0.68 | 260 | 1.29 | 252 | 2.53 | 299 | 0.49 | 316 | 0.25 | 248 | 0.12 | 338 | 4.07 | 185 |

## 3. Materials and Methods

Using the latest data from Tehran municipality, the Statistical Center of Iran, and Landsat satellite images, this paper aims to improve land use efficiency evaluation and land consumption frameworks (Figure 3).

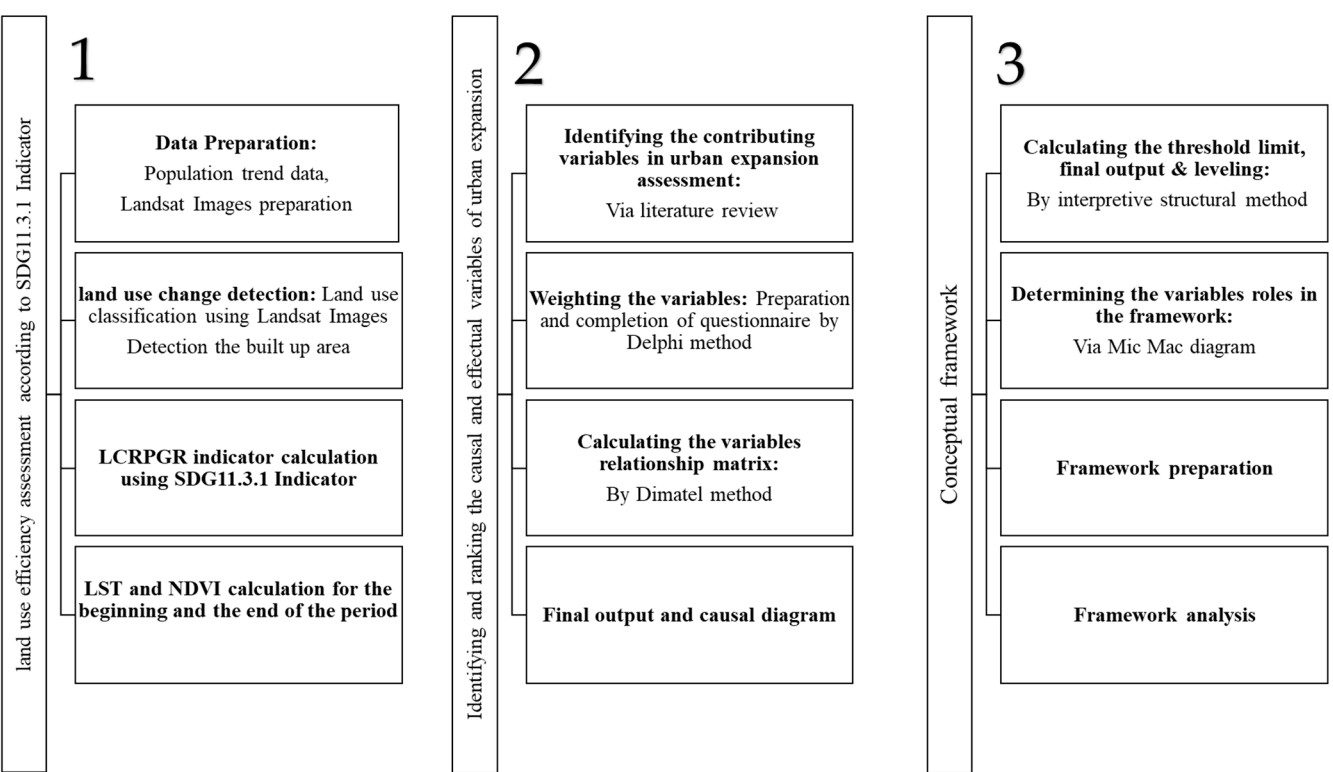

**Figure 3.** Research phases.

*3.1. Tehran's Land Use Efficiency Assessment According to SDG 11.3.1 Indicator*

The first phase involves assessing Tehran's land use efficiency using the SDG 11.3.1 Indicator. In order to calculate this indicator for Tehran, official census data and satellite imagery were used to classify land cover (Table 3). With a 5-year interval, the study period runs from 1986 to 2021. The reason for selecting 1986 as the starting year is to highlight the importance of the urban development process in Iran, along with the availability of satellite images with the appropriate resolution and thermal band. In 1986, Iran's urbanization rate exceeded 50% for the first time, reaching 54%, according to the Iran Statistics Center. This period coincided with the formation of various formal and informal settlements around the city of Tehran and its rapid expansion.

**Table 3.** The characteristics of images derived from satellites.

| Image | Source | Date | Sensor | Overall Accuracy | Kappa Co., |
|-------|--------|------|--------|------------------|-----------|
| Landsat | USGS | 1986/07/14 | L5_TM | 89.4 | 0.9 |
| Landsat | USGS | 1991/07/18 | L5_TM | 88.8 | 0.89 |
| Landsat | USGS | 1996/07/15 | L5_TM | 91.2 | 0.91 |
| Landsat | USGS | 2001/07/18 | L5_TM | 93.6 | 0.93 |
| Landsat | USGS | 2006/08/01 | L5_TM | 92.7 | 0.91 |
| Landsat | USGS | 2011/07/09 | L5_TM | 91.8 | 0.9 |
| Landsat | USGS | 2016/07/22 | LANDSAT_8 | 93.1 | 0.92 |
| Landsat | USGS | 2021/07/20 | LANDSAT_8 | 91.9 | 0.91 |

3.1.1. Landsat Images Land Cover Classification

A series of satellite images were analyzed to determine Tehran's land cover. Using ENVI software, a supervised neural network was developed for eliciting land cover classes after pre-processing and modifying image errors. This method was employed to categorize the images into four categories: barren land, vegetation, water, and built-up areas. The algorithm used for training was the backpropagation algorithm, the training rate was 0.9, and five hidden layers and 1000 iterations were performed. The classification was completed

using the elicited training data from satellite images with high accuracy and municipal land use data of Tehran, and 70% of the samples were considered as training samples and 30% for testing. Classification accuracy was tested by examining the overall accuracy and the Kappa coefficient. The overall accuracy is an average of classification accuracy, which shows the ratio of correctly classified pixels to the total number of known pixels. The Kappa coefficient also determines classification accuracy when compared to completely random classification. It is usually stated that the overall accuracy is an optimistic estimate and always calculates the accuracy higher than the actual value (overestimation), whereas the Kappa coefficient is a pessimistic estimate and expresses the accuracy lower than the actual value (underestimation).

### 3.1.2. LST and NDVI Calculation

Urban expansion is believed to have the greatest environmental impact by reducing vegetation, increasing surface temperatures, and forming urban heat islands. Moreover, the lack of urban services is one of the biggest challenges to urban livability [34–38]. Thus, this paper aims to examine how land use changes impact ecosystem services by calculating Land Surface Temperature (LST) and Normalized Difference Vegetation Index (NDVI) over the selected period (1986–2021). LST was calculated using a Split-Window Algorithm (SWA), and NDVI was determined using the threshold limit method [39].

### 3.1.3. LCRPGR Indicator Calculation

Then, using prepared data, the land use efficiency indicator based on the 11.3.1 indicator has been calculated using Equation (1). LCR represents land use changes in two successive periods, and PGR represents demographic changes in two successive periods. As can be seen in Equations (2) and (3), LCR and PGR are obtained from the Ln of value division of the area of the built environment (Urb) and the population (Pop) of the most recent year (t + n) to that of the former year (t) divided by y, representing the number of years between them.

$$LCRPGR = \frac{LCR}{PGR} \tag{1}$$

$$LCR = \frac{\mathrm{Ln}\left(\frac{Urb\ (t+n)}{Urb\ t}\right)}{y} \tag{2}$$

$$PGR = \frac{\mathrm{Ln}\left(\frac{Pop\ (t+n)}{Pop\ t}\right)}{y} \tag{3}$$

### 3.2. Identifying and Ranking the Casual and Effectual Variables of Urban Expansion

To develop and improve the land use efficiency assessment indicator, the DEMATEL method was used to identify and rank the casual and effectual contributing variables associated with urban expansion, including ecosystem services, urban environmental capacities, urban services, and livability. DEMATEL is a multi-criteria decision-making method based on directed graphs that divide variables into two groups of cause and effect and illustrate the relationship between the main variables of a system, the number of relationships, and the extent to which the main variables influence and are influenced by them [40,41]. DEMATEL is used when solving complicated problems and analyzing cause-and-effect relationships between variables with a low degree of certainty. It aims to present a conceptual framework and a model of different and sometimes opposite subjects. Modeling with fuzzy DEMATEL is a form of modeling in an uncertain scenario that employs the opinion of experts on how to organize structures through scoring fuzzy numbers. This is one of the methods used to make decisions based on paired samples. Experts' ideas are used to extract the parameters of a system, and then graph theory is used to structure them. It presents a hierarchical structure and scores existing variables in systems with cause–effect relations. Additionally, the DEMATEL method is used for identifying and

analyzing the mutual relationships between the criteria and for creating relation networks. Due to the fact that directed graphs can better show the relationships between elements, the DEMATEL method is based on charts that divide variables into two groups of cause and effect and turn these relationships into a comprehensible structural model [40].

### 3.2.1. Identification of Variables Affecting and/or Being Affected by Urban Expansion

First, through a content analysis of the related literature, ten mainly repeated variables were collected in the three categories of environmental, physical–spatial, and socioeconomic effects of urban expansion (Table 4). The environmental category includes green spaces, water and air quality, and sewage and waste management variables. Transport networks, infill development, and the spatial distribution of urban services are included in the physical–spatial category. The socioeconomic category consists of entrepreneurship and housing and population attraction variables. Population attraction variable refers to the capacity for population attraction that is created by housing production as an incentive for immigrants to move to Tehran.

**Table 4.** Land use efficiency assessment variables based on the carrying capacity of the environment.

| Categories | Variables | Sources |
|---|---|---|
| Environmental | Green spaces and vegetation<br>The required water sources for the city residents and other urban consumptions<br>Air quality (air quality index/amount of air pollution)<br>The amount of producing, gathering and recycling of sewage and waste | [41–60] |
| Physical–spatial | The quantity and quality of road networks, traffic infrastructures and public transportation<br>Infill development<br>Spatial justice | [41,55,57,58,60] |
| Socioeconomic | Housing and population attraction<br>Entrepreneurship | [41,46,47,51,52,57,60] |

### 3.2.2. Preparation and Completion of a Questionnaire Using the Delphi Method

Following that, two main research questions were answered using a questionnaire based on these variables. The questions were designed to reveal causal relations between variables and also classify them by 14 experts engaged in urban planning, architecture, urban climatology, urban policy makers, urban eco-environmental studies, and urban socioeconomic studies. The experts were selected from reputable professionals and university professors in Iran. As the issues included in the questionnaire are the results of research with deep content analysis techniques that have been approved by Iranian professors of urban planning, it has high validity (Table 5).

**Table 5.** Variables used in the assessment.

| Number | Variables | ID | Number | Variables | ID |
|---|---|---|---|---|---|
| 1 | Land surface temperature | A1 | 6 | Transportation | A6 |
| 2 | Air pollution | A2 | 7 | Infill development | A7 |
| 3 | Sewage and waste | A3 | 8 | Spatial justice | A8 |
| 4 | Water resources | A4 | 9 | Housing and population attraction | A9 |
| 5 | Vegetation | A5 | 10 | Entrepreneurship | A10 |

### 3.2.3. Scoring Method

Fuzzy numbers were used in this research. Table 6 shows the scoring range and the associated statements.

**Table 6.** Statements and the way of scoring using fuzzy method [61].

| Code | Statement | L | M | U |
|------|-----------|---|---|---|
| 1 | No impact | 0 | 0 | 0.25 |
| 2 | Very low impact | 0 | 0.25 | 0.5 |
| 3 | Low impact | 0.25 | 0.5 | 0.75 |
| 4 | High impact | 0.5 | 0.75 | 1 |
| 5 | Very high impact | 0.75 | 1 | 1 |

3.2.4. Variables Relationship Matrix by DEMATEL Method

- Formation of fuzzy direct correlation matrix

In order to identify the pattern of relationship among *n* criteria, first a *n\*n* matrix was formed (Equation (4)). The effect of elements in each row on the elements in each column was inserted as a fuzzy number in this matrix. If the opinion of more than one expert is used, each of the experts should complete the existing matrix. The output of this phase is a direct correlation matrix (z) or pair comparisons of the expert's comments. In cases where more than one expert is used in the evaluation, this matrix represents the arithmetic mean of all the experts' opinions. In the next phase, the existing relationships between variables will be normalized in order to make the numbers in the output table of this phase (Table A1 in the Appendix A) comparable and evaluable.

$$z = \begin{bmatrix} 0 & \cdots & \widetilde{z}_{n1} \\ \vdots & \ddots & \vdots \\ \widetilde{z}_{1n} & \cdots & 0 \end{bmatrix} \tag{4}$$

- Normalizing the fuzzy direct correlation matrix

The following equation was used for normalizing fuzzy direct correlation matrix.

$$\widetilde{x}_{ij} = \frac{\widetilde{z}_{ij}}{r} = \left( \frac{l_{ij}}{r}, \frac{m_{ij}}{r}, \frac{u_{ij}}{r} \right) \tag{5}$$

$$r = \max_{i,j} \left\{ \max_i \sum_{j=1}^n u_{ij}, \max_j \sum_{i=1}^n u_{ij} \right\} \quad i,j \in \{1,2,3,\ldots,n\} \tag{6}$$

- Calculation of the complete correlation fuzzy matrix

In this step, the total fuzzy correlation matrix was formed using the following equation. The inverse of the normal matrix was first calculated, then subtracted from matrix I, then multiplied by the resulting matrix. Table A2 in Appendix A shows the complete fuzzy correlation matrix.

$$\widetilde{T} = \lim_{k \to +\infty} \left( \widetilde{x}^1 \oplus \widetilde{x}^2 \oplus \ldots \oplus \widetilde{x}^k \right) \tag{7}$$

$$\widetilde{t}_{ij} = \left( 1''_{ij}, m''_{ij}, u''_{ij} \right) \tag{8}$$

$$\left[ l''_{ij} \right] = x_l \times (I - x_l)^{-1} \tag{9}$$

$$\left[ m''_{ij} \right] = x_m \times (I - x_m)^{-1} \tag{10}$$

$$[u''_{ij}] = x_u \times (I - x_u)^{-1} \tag{11}$$

- De-fuzzification of the complete correlation matrix

De-fuzzification was performed using the Converting Fuzzy data into Crisp Scores (CFCS) method developed by Opricovic and Tzeng [62]. De-fuzzified correlation matrix

values are shown in Appendix A, Table A3. The stages of the de-fuzzification method are as follows:

$$l_{ij}^n = \frac{\left(l_{ij}^t - \min l_{ij}^t\right)}{\Delta_{min}^{max}} \tag{12}$$

$$m_{ij}^n = \frac{\left(m_{ij}^t - \min l_{ij}^t\right)}{\Delta_{min}^{max}} \tag{13}$$

$$u_{ij}^n = \frac{\left(u_{ij}^t - \min l_{ij}^t\right)}{\Delta_{min}^{max}} \tag{14}$$

$$\Delta_{min}^{max} = \max u_{ij}^t - \min l_{ij}^t \tag{15}$$

$$l_{ij}^s = m_{ij}^n / \left(1 + m_{ij}^n - l_{ij}^n\right) \tag{16}$$

$$u_{ij}^s = u_{ij}^n / (1 + u_{ij}^n - l_{ij}^n) \tag{17}$$

The output of the CFCS algorithm is a matrix with certain amounts. Equation 15, 16, and 17 were used to calculate the total normalized certain amounts.

- Final output and casual diagram

The next step is to sum the rows and columns of the T matrix shown in Appendix A, Table A4. Following are the equations for calculating the sum of rows (D) and columns (R).

$$D = \sum_{j=1}^n T_{ij} \tag{18}$$

$$R = \sum_{i=1}^n \widetilde{T}_{ij} \tag{19}$$

A variable's effect on other variables in the system is represented by R, which is the sum of the elements in each row (the variable's effectiveness). The sum of the elements of each column (D) shows the effectiveness of other variables on that variable (the extent to which it has been affected). These are calculated in this way, and then we calculate the amounts of D + R and D − R. D + R indicates the amount that the intended variable affects and is being affected. This means that the higher the D + R, the greater the interaction between that variable and others. D-R reflects the power of each variable's effectiveness. A positive index indicates that the variable is causal, whereas a negative index indicates that it is effectual. Finally, a Cartesian coordinate system is drawn. In this system, the horizontal axis is related to D + R and the vertical axis is related to D-R. A point determines the position of each variable with the coordinates (D − R, D + R) in the system.

### 3.3. Conceptual Framework

3.3.1. Obtaining Threshold Limits, Final Output and Levels Using the Interpretive Structural Modeling Method

To determine the impact of each variable on land use efficiency, Interpretive Structural Modeling (ISM) was used following DEMATEL analysis. DEMATEL's final output was used in this phase and certain relationships were achieved by removing the threshold limit. Interpretive Structural Modeling methods are complementary when non-quantitative variables need to be leveled.

3.3.2. Determination of the Variables' Roles in the Framework

Having determined the levels of the variables, a general framework and conceptual diagram were prepared to assess the meaningful statistical and descriptive relationships between the variables. The analysis of these relationships was conducted via Micmac software.

## 4. Results and Discussion

### 4.1. Tehran's Land Use Efficiency Evaluation According to SDG 11.3.1 Indicator

Tehran is the capital city and the most crowded city in Iran. According to the country's latest official census, while the country's population growth rate equals 1.24, Tehran province's rate is 1.78 [63]. As Tehran received the most immigrants during the previous statistical period, it now experiences an inverse immigration process and is the destination of reverse immigration from Tehran as well as immigration from other regions of the country to Tehran province in the latest statistical period. The reason is the high rate of residency in Tehran. Both groups of these immigrants have a job in Tehran but cannot afford a house in this city. That is why this issue can make a big part of the natural lands of the surrounding cities of Tehran be used in construction and lead to spot growth due to the metropolitan effects of Tehran.

#### 4.1.1. Land Cover Classification Using Landsat Satellite Images

The land cover of Tehran was determined using satellite images. In Table 7, built-up areas, barren land, vegetation, and water over 35 years are shown based on an interval of five years. Land has primarily been used for construction as built-up areas have risen, while other land cover classes have decreased, especially barren land. Changes in land cover in Tehran indicate that between 1991 and 1996, the highest increase in built-up areas took place. Land prices were very low during this time, and immigration policies (another motivating factor was the presence of land and housing) led to a population explosion. In the so-called Construction Period, which began just after the Iran–Iraq War due to the large number of reconstructions and economic revolutions that took place after the war, political economy encouraged high construction, whose effects can also be seen in the following periods.

**Table 7.** A comparison of the four land cover areas in Tehran between 1986 and 2021.

| Year | Built Up | | Water | | Barren Land | | Vegetation | |
|------|----------|--|-------|--|-------------|--|------------|--|
| | Area (ha) | % Of Total Area | Area (ha) | % Of Total Area | Area (ha) | % Of Total Area | Area (ha) | % Of Total Area |
| 1986 | 31,730.83 | 52% | 22.77 | 0.04% | 20,592.08 | 33% | 9216.2473 | 15% |
| 1991 | 33,900.43 | 55% | 21.42 | 0.03% | 19,002 | 31% | 8638.078 | 14% |
| 1996 | 40,935.6 | 66% | 20.88 | 0.03% | 14,235.85 | 23% | 6369.597 | 10% |
| 2001 | 41,810.81 | 68% | 21.24 | 0.03% | 12,290.13 | 20% | 7439.7445 | 12% |
| 2006 | 44,981.29 | 73% | 20.34 | 0.03% | 9695.328 | 16% | 6864.9672 | 11% |
| 2011 | 46,082.84 | 75% | 20.36 | 0.03% | 9091.11 | 15% | 6367.6166 | 10% |
| 2016 | 46,663.47 | 76% | 113.13 | 0.18% | 6653.662 | 11% | 8131.6589 | 13% |

In 2016, the Chitgar artificial lake was constructed in District 22 of Tehran, causing most of the changes in water bodies. The majority of changes in barren lands occurred between 1991 and 1996, and the majority of these lands have been used for residential and commercial construction purposes. The areas of vegetation and green spaces have not followed a regular trend and have undergone slight changes depending on urban management policies. Although Tehran Municipality intended to increase this area in Tehran, the total area has remained the same over the studied period. From 2011 to 2016, vegetated areas increased as the municipality's green space was planted and urban beautification policies were implemented. As a result of these policies, the margins of highways have been filled with green spaces, river valleys have been reorganized, and several major and city-scale parks have been built. The trend of these changes can be seen in Figure 4.

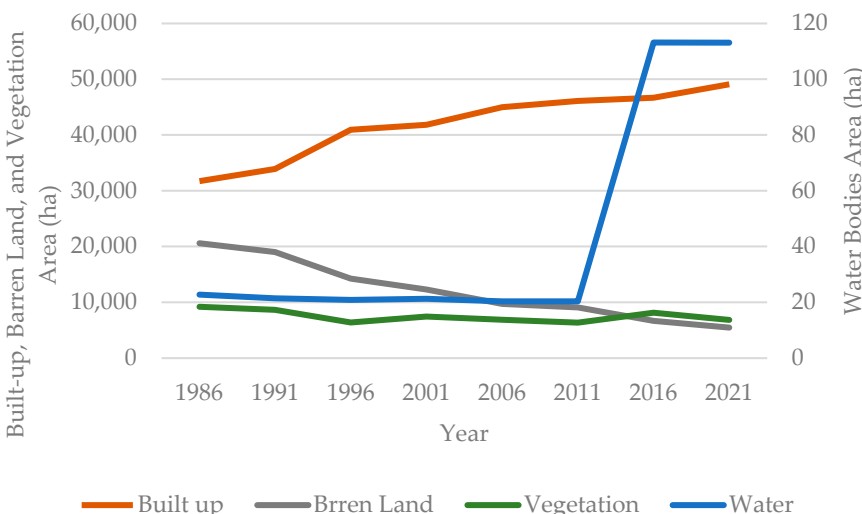

**Figure 4.** Changing trends in Tehran's four land cover areas.

Table 8 shows how much land cover has changed by class. Built-up areas have been increasing while barren lands have been decreasing, indicating that most of the changes have been from barren lands to built-up areas. Even though the municipality has developed urban services such as green spaces and artificial lakes during this period, the general trend of land cover changes shows that natural lands have been predominantly converted into built-up areas (Figure 5).

To discuss the above-mentioned results, it can be argued that one of the critical urban policies which is among the underlying reasons for such an urban expansion in Tehran dates back to 1991–1996, when Districts 21 and 22 were added to Tehran's legal boundary through a new Master Plan. In this plan, the majority of the land in these western districts was dedicated to various urban services and green areas accompanied by low-density residential areas. While these districts were experiencing normal growth, from 2001 onwards, the spatial structure and population of these two districts dramatically changed due to a new urban growth policy. Tehran Municipality changed the Master Plan's land use after these districts became legal districts for a variety of reasons, including financial gain, and dedicated a significant portion of these districts, particularly District 22, to high-rise, densely populated residential complexes and mega malls mostly constructed by governmental organizations, banks, and major cooperatives.

**Table 8.** Tehran's land cover changes within the study period.

|  | 1986–1991 | | 1991–1996 | | 1996–2001 | | 2001–2006 | |
|---|---|---|---|---|---|---|---|---|
|  | ha | % | ha | % | ha | % | ha | % |
| Built up | 2169.6 | 6.8 | 7035.2 | 20.8 | 875.2 | 2.1 | 3170.5 | 7.6 |
| Water | −1.4 | −5.9 | −0.5 | −2.5 | 0.4 | 1.7 | −0.9 | −4.2 |
| Baren Land | −1590.1 | −7.7 | −4766.2 | −25.1 | −1945.7 | −13.7 | −2594.8 | −21.1 |
| Vegetation | −578.2 | −6.3 | −2268.5 | −26.3 | 1070.1 | 16.8 | −574.8 | −7.7 |
|  | **2006–2011** | | **2011–2016** | | **2016–2021** | | **1986–2021** | |
|  | ha | % | ha | % | ha | % | ha | % |
| Built up | 1101.5 | 2.4 | 580.6 | 1.3 | 2413.8 | 5.2 | 17346.4 | 54.7 |
| Water | 0.0 | 0.1 | 92.8 | 455.6 | −0.1 | −0.1 | 90.29 | 396.5 |
| Baren Land | −604.2 | −6.2 | −2437.4 | −26.8 | −1165.9 | −17.5 | −15104 | −73.4 |
| Vegetation | −497.4 | −7.2 | 1764.0 | 27.7 | −1289.8 | −15.9 | −2374.4 | −25.8 |

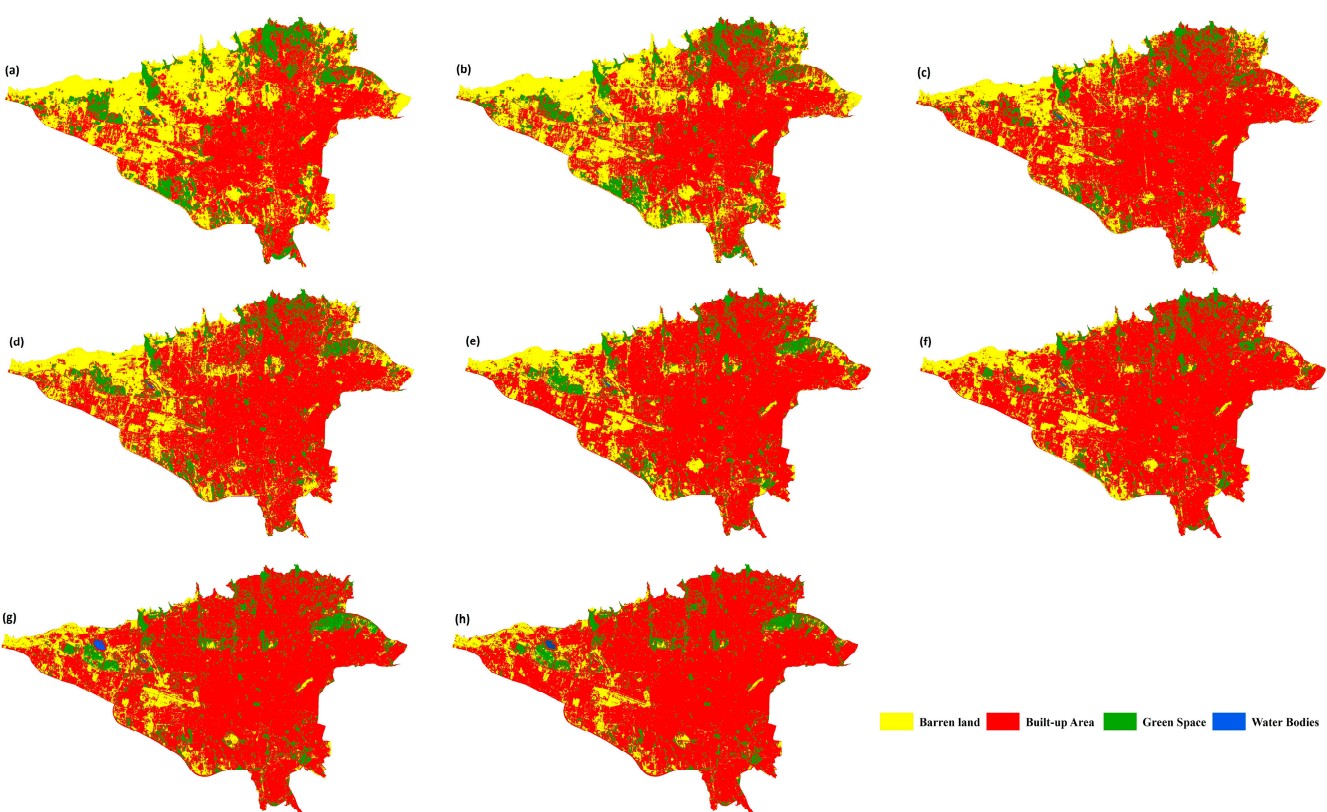

**Figure 5.** Tehran land cover changes between 1986 and 2021: (**a**) 1986, (**b**) 1991, (**c**) 1996, (**d**) 2001, (**e**) 2006, (**f**) 2011, (**g**) 2016, and (**h**) 2021.

### 4.1.2. LCRPGR Indicator for Tehran

Tehran's LCRPGR were calculated by taking into account land cover changes, built-up areas, and population changes. Table 9 shows how different types of urban expansion are interpreted using LCRPGR. According to this indicator, Tehran has never experienced urban shrinkage. This indicates that this city is experiencing a growing trend. During four periods, including 1996–2001, 2001–2006, 2006–2011, and 2011–2016, the amount of LCRPGR indicates that population growth exceeded land consumption. However, during three periods of 1986–1991, 1991–1996, and 2016–2021, we faced rapid urban expansion, which means that the land consumption rate exceeded the population growth rate.

**Table 9.** Different kinds of urban expansion according to LCRPGR indicator [64].

| Urban Expansion Type | LCRPGR | Interpretation |
|---|---|---|
| Rapid Urban Growth | Between 1 and 5 | Urban land consumption exceeds population growth |
| Rapid Urban Population Growth | Between 0 and 1 | Urban population growth exceeds land consumption |
| Urban Shrinking | Between −5 and 0 | Urban population declining or urban land shrinking |

Irregular changes in LCRPGR can indicate political changes or a lack of territory planning. In the recent period of 2016 to 2021, the city of Tehran experienced expansion and an increase in the built-up area again. Investigating the three recent statistical periods, which coincide with the three recent periods of this paper, shows that this increase has not led to a better provision of services per capita and more access of neighborhoods to urban facilities. Figure 6 shows the process of changes in eight urban services per capita in three statistical periods.

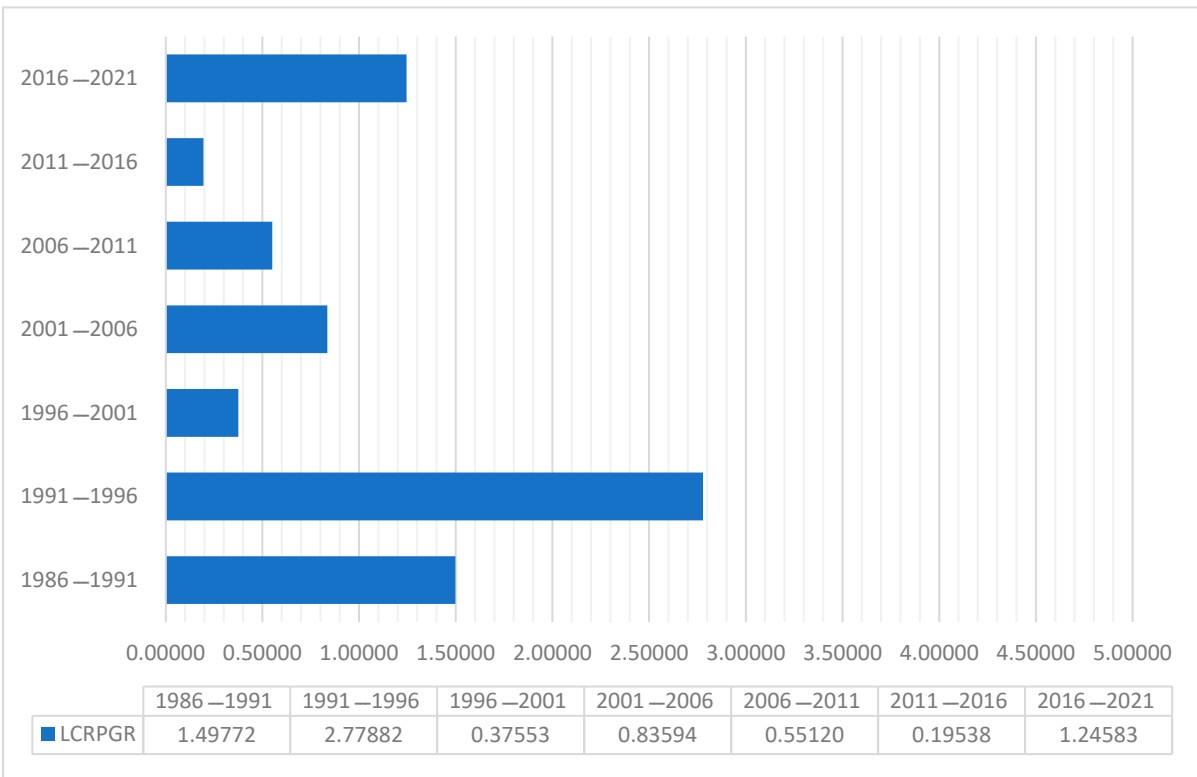

| | 1986—1991 | 1991—1996 | 1996—2001 | 2001—2006 | 2006—2011 | 2011—2016 | 2016—2021 |
|---|---|---|---|---|---|---|---|
| ■ LCRPGR | 1.49772 | 2.77882 | 0.37553 | 0.83594 | 0.55120 | 0.19538 | 1.24583 |

**Figure 6.** Diagram of land use efficiency based on LCRPGR.

Figure 7 shows changes in urban services per capita. Other than changes in green space per capita, other services per capita have decreased or remained unchanged. Specifically, with changes in population and urban divisions, an increase in the number of neighborhoods, and their uneven distribution in different districts and neighborhoods, the increase in green spaces per capita has not meaningfully improved the level of access to these areas for local communities.

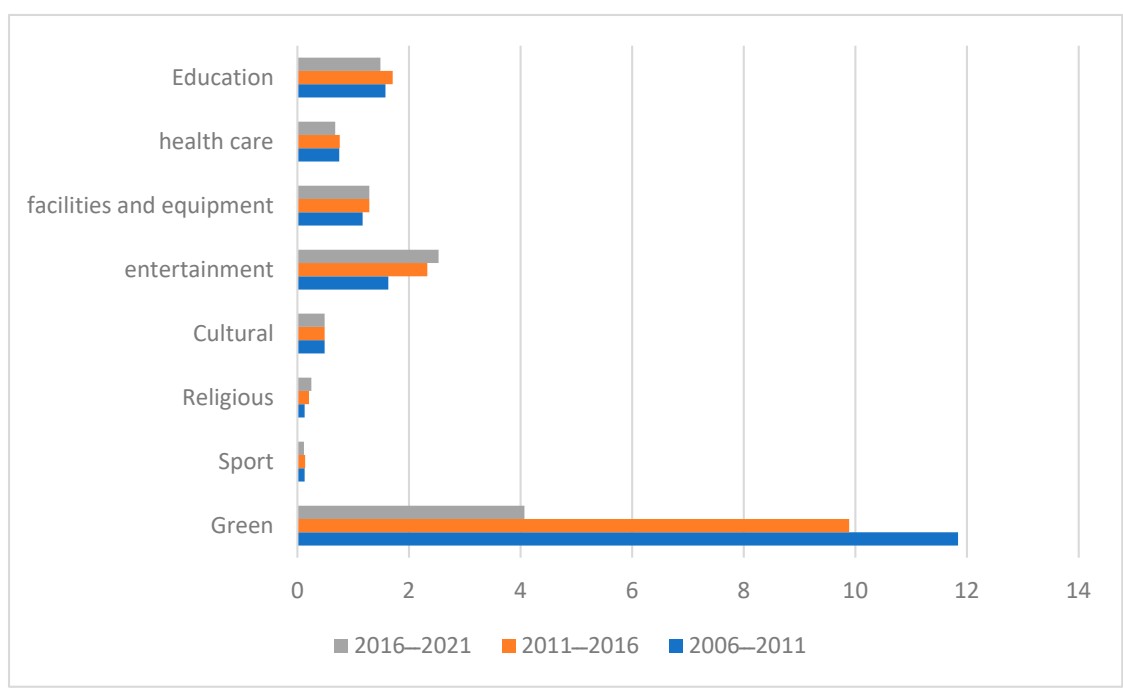

**Figure 7.** Changes in eight urban services per capita.

As a general rule, city expansion should provide residents with better services and improve their quality of life. However, Figure 8 illustrates that in the case of Tehran, the number of neighborhoods with inadequate urban services does not seem to be declining during the studied period. While the number of neighborhoods lacking green spaces and religious and educational facilities per capita has increased, this measure for other urban services has remained relatively unchanged.

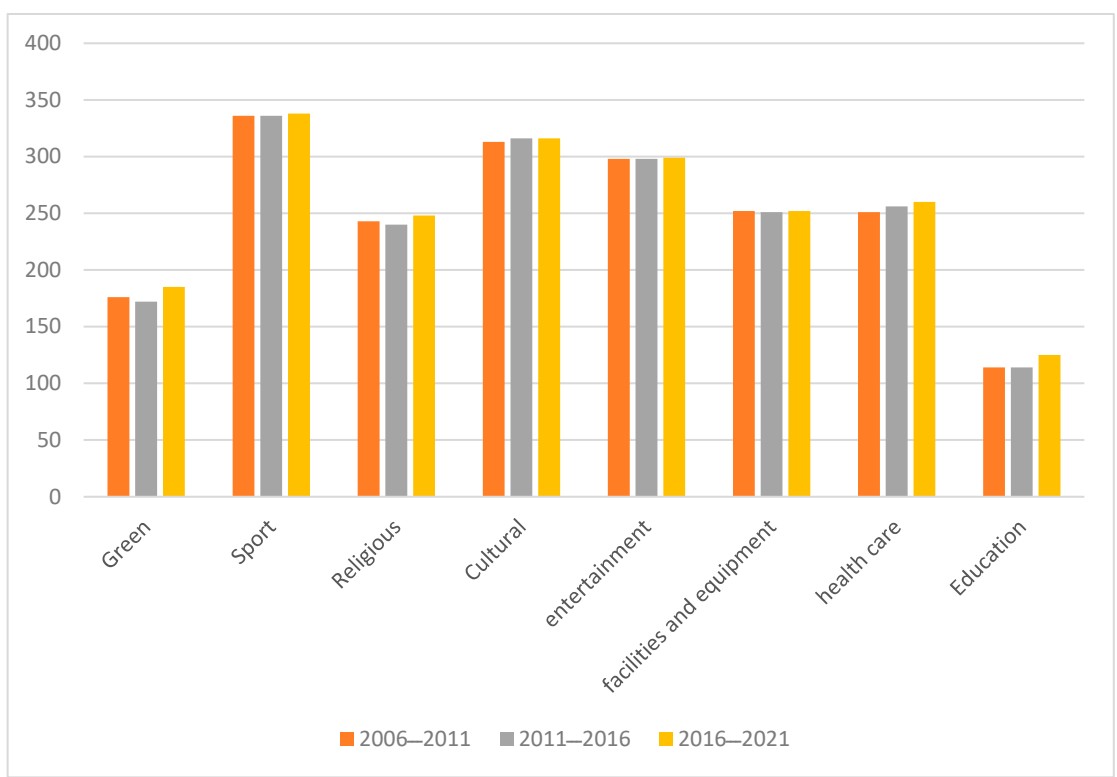

**Figure 8.** The number of neighborhoods with a per capita shortage of eight urban services.

### 4.1.3. LST and NDVI

Growth in built-up areas, along with changes in land surface temperature (LST) and the normalized index of difference in vegetation (NDVI), indicates a reduction in vegetation and an increase in surface temperature (Figure 9). In fact, as the city's population density and built-up area increased, ecosystem services from urban green spaces decreased, while average surface temperatures increased during the study period. As shown in Table 10, comparing the minimum, average, and maximum temperatures as well as the NDVI between the first and last years of the study shows a significant increase in LST and a meaningful decrease in NDVI.

**Table 10.** Changes in temperature and vegetation index.

| LST | Min (°C) | Max (°C) | Average (°C) |
|---|---|---|---|
| 1986 | 16.56 | 39.33 | 27.945 |
| 2021 | 24.49 | 47.25 | 35.87 |
| **NDVI** | **Min** | **Max** | **Average** |
| 1986 | −0.526 | 0.983 | 0.2285 |
| 2021 | −0.57 | 0.579 | 0.2045 |

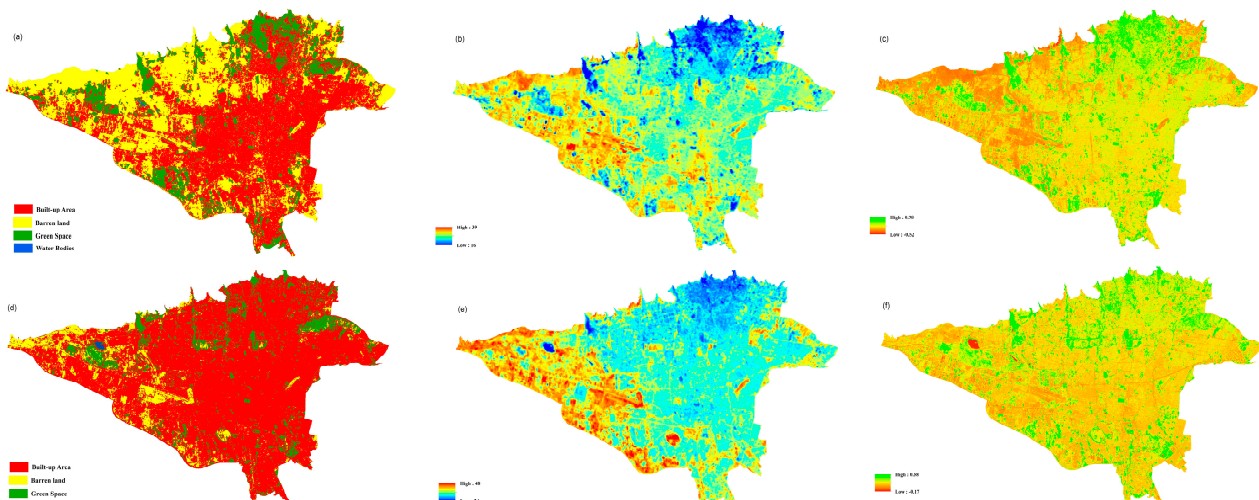

**Figure 9.** Tehran land cover 1986 (**a**), Tehran LSt 1986 (**b**), Tehran NDVI 1986 (**c**) and Tehran land cover 2021 (**d**), Tehran LSt 2021 (**e**), Tehran NDVI 2021 (**f**).

*4.2. DEMATEL Results*

By implementing a DEMATEL model on the obtained data, causal relationships between variables were discovered. As was fully explained in the methodology section, the fuzzy direct correlation was first calculated, and then the fuzzy direct correlation matrix was normalized, the fuzzy correlation matrix was calculated, de-fuzzification of the values of the complete correlation matrix was performed, and the threshold limit was calculated, and, accordingly, the final output of the model is shown in Table 11. The values of D + R and D − R indicate the amount of interaction and amount of effectiveness of the variables, respectively.

**Table 11.** Final output of DEMATEL.

| Variables | R | D | D − R | D + R |
|:---:|:---:|:---:|:---:|:---:|
| A1 | 4.196 | 3.936 | −0.26 | 8.131 |
| A2 | 4.239 | 3.995 | −0.245 | 8.234 |
| A3 | 3.991 | 3.761 | −0.231 | 7.752 |
| A4 | 4.219 | 4.105 | −0.114 | 8.324 |
| A5 | 4.233 | 4.086 | −0.147 | 8.32 |
| A6 | 3.821 | 4.213 | 0.393 | 8.034 |
| A7 | 3.997 | 4.228 | 0.231 | 8.225 |
| A8 | 4.054 | 3.958 | −0.097 | 8.012 |
| A9 | 4.235 | 4.462 | 0.227 | 8.697 |
| A10 | 3.769 | 4.012 | 0.243 | 7.781 |

Figure 10 shows the pattern of meaningful cause–effect relations. In this figure, the horizontal axis represents the amount of D+R and the vertical axis indicates D − R. The position and relations of each variable are determined by a point (D + R, D − R) in the coordinate system.

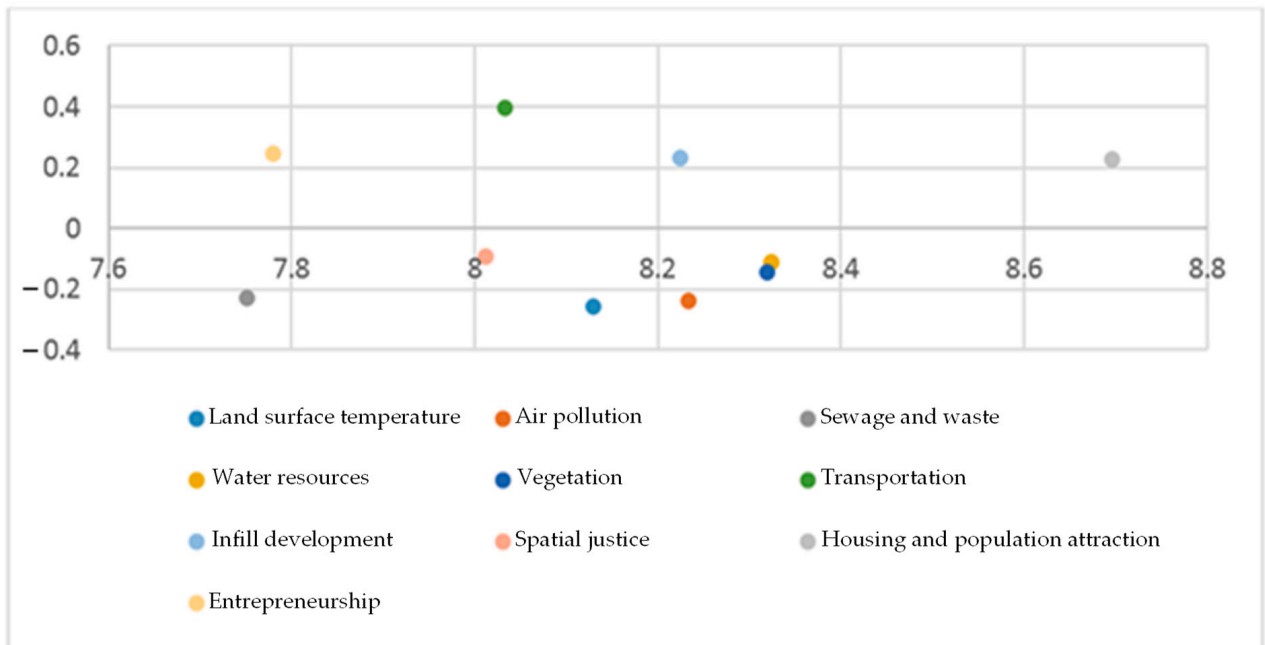

**Figure 10.** Cause–effect diagram.

There are four aspects taken into account when analyzing the variables in Figure 10 and Table 11:

1. The degree of variables' effectiveness: The sum of the elements of each row (D) for each variable indicates the amount of that variable's effectiveness on other variables of the system.

2. The number of variables being affected: The sum of the elements of each column (R) for each variable indicates the amount that variable is being influenced by other elements of the system. As a result of this study, air pollution is more affected than any other variable, followed by housing and population attraction, vegetation, water resources, land-surface temperature, spatial justice, infill development, sewage and waste, transportation, and entrepreneurship.

3. The horizontal axis (D + R) represents the degree to which the intended variables affect and are affected by the system. Therefore, the higher the D+R of a variable, the greater the interaction between that and other variables in the system. According to this study, housing and population attraction are the most effective variables, followed by water, vegetation, air pollution, infill development, land-surface temperature, transportation, spatial justice, entrepreneurship, and sewage and waste.

4. Each variable's effectiveness is expressed on the vertical axis (D−R). Generally, if D−R is positive, the variable is considered a cause and if it is negative, it is viewed as an effect. As part of this study, transportation, infill development, housing and population attraction, and entrepreneurship are considered causes, while air pollution, sewage and waste, water resources, vegetation, and spatial justice are considered effects.

Following DEMATEL analysis, in order to determine the levels of each of the variables and the level being affected, the Interpretive Structural Modeling (ISM) method was used. Using this method, a certain relationship matrix is created by removing a threshold amount (Table 12).

**Table 12.** Relationship matrix obtained by removing the threshold value using the ISM method.

| Variables | A1 | A2 | A3 | A4 | A5 | A6 | A7 | A8 | A9 | A10 | D Values |
|---|---|---|---|---|---|---|---|---|---|---|---|
| A1 | 0 | 1 | 0 | 1 | 1 | 0 | 0 | 0 | 1 | 0 | 4 |
| A2 | 1 | 0 | 0 | 1 | 1 | 0 | 0 | 0 | 1 | 0 | 4 |
| A3 | 0 | 0 | 0 | 1 | 1 | 0 | 0 | 0 | 0 | 0 | 2 |
| A4 | 1 | 1 | 1 | 0 | 1 | 0 | 1 | 1 | 1 | 0 | 7 |
| A5 | 1 | 1 | 1 | 1 | 0 | 0 | 0 | 1 | 1 | 0 | 6 |
| A6 | 1 | 1 | 1 | 1 | 1 | 0 | 1 | 1 | 1 | 0 | 8 |
| A7 | 1 | 1 | 1 | 1 | 1 | 1 | 0 | 1 | 1 | 0 | 8 |
| A8 | 1 | 1 | 0 | 1 | 1 | 0 | 0 | 0 | 1 | 0 | 5 |
| A9 | 1 | 1 | 1 | 1 | 1 | 1 | 1 | 1 | 0 | 1 | 9 |
| A10 | 1 | 1 | 0 | 1 | 0 | 0 | 0 | 1 | 1 | 0 | 5 |
| R values | 8 | 8 | 5 | 9 | 8 | 2 | 3 | 6 | 8 | 1 | - |

Each column in the matrix shown in Table 12 indicates the level of variables' dependence on each other. Thus, the most dependent variable is water resources (A4) followed by land-surface temperature (A1), air pollution (A2), vegetation (A5), and housing and population attraction (A9). The entrepreneurship variable (A10) has the least dependence. Each row indicates the level of impact one specific variable has on the other variables. Therefore, housing and population attraction (A9), infill development (A7), and transportation (A6) are the variables with the highest level of effectiveness, respectively. The Sewage and waste variable (A3) has the lowest level of effectiveness on other variables.

*4.3. Land Use Efficiency Evaluation Framework*

The results derived from the matrix in Table 12 are summarized in Table 13. According to the values obtained for the effectiveness and affectability of each variable, the Micmac diagram (Figure 11) was developed to determine the final level of each variable.

**Table 13.** Variables' level, effectiveness and affectability.

| Variables | Level | Effectiveness | Affectability |
|---|---|---|---|
| A1 | 1 | 4 | 8 |
| A2 | 1 | 4 | 8 |
| A3 | 1 | 2 | 5 |
| A4 | 1 | 7 | 9 |
| A5 | 1 | 6 | 8 |
| A6 | 3 | 8 | 2 |
| A7 | 3 | 8 | 3 |
| A8 | 2 | 5 | 6 |
| A9 | 2 | 9 | 8 |
| A10 | 3 | 5 | 1 |

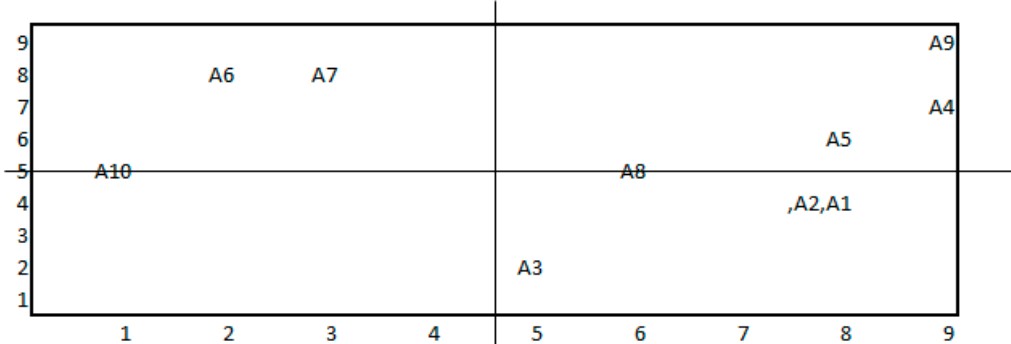

**Figure 11.** Using the Micmac diagram to determine variables' levels.

Figure 12 presents the land use efficiency evaluation framework derived from Table 13 and Figure 11. As can be seen in this figure, variables are classified into three levels. Transportation (A6), infill development (A7), and entrepreneurship (A10) variables can be regarded as independent variables. These variables are the most important and have the highest effectiveness on other variables and the urban expansion of Tehran. They are, in fact, the effective variables of the system, and they are less affected by other variables. It should be mentioned that in this research, entrepreneurship can be seen as an autonomous variable. Considering that this variable has the lowest correlation with the general structure of the variables, it should be excluded from the structural analysis. Land-surface temperature (A1), air pollution (A2), sewage and waste (A3), water resources (A4), and vegetation (A5) can be considered dependent variables. These variables have the lowest effectiveness and are most affected by other variables. Spatial justice (A8) and housing and population attraction (A9) are the linking variables that transmit the effect of independent variables and act as the system's central factors. This means that these middle-level variables simultaneously affect the first-level variables and are being affected by the third-level variables.

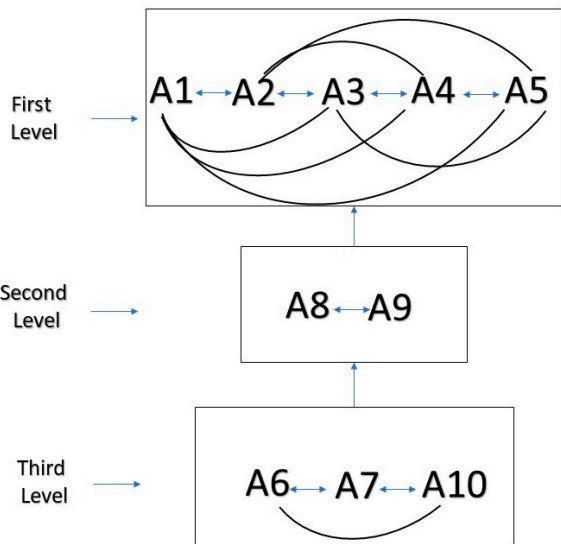

**Figure 12.** Land use efficiency planning and evaluation framework.

We have several findings about the way the three variables of transportation, infill development and entrepreneurship influence urban expansion align with others. For instance, Zhao et al. [65] in their study focused on the importance of transportation in relation to urban expansion. They argue that urban form and commuting patterns are closely related. In fact, all types of land use have economic and social characteristics when distance and transportation costs are considered. The frequency and length of commuting are increased by low-density and sprawling development patterns, while they are decreased by compact and infill urban development. It is imperative to admit that urban expansion is affected not only by transportation systems and urban forms but also by the market power and performance of landowners, banks, and private entities [66], which have led to dispersed urban cores globally [67].

The spatial justice and housing and population attraction variables shape the middle level. In most studies, spatial justice is considered an important geographical analysis in planning, despite not having a strong predetermined framework yet [68]. This concept, which originated in 1960s and 1970s studies, forms the basis of every spatial analysis of a city [69,70]. In fact, maintaining a well-developed transportation network or implementing an infill development policy in deprived areas and brownfields can contribute to spatial justice, and at the same time, spatial injustice can exacerbate lack of access to urban resources, green spaces, among others. Another middle-level variable, housing and population attraction, is affected by third-level variables and also affects first-level variables.

It is considered to be one of the most fundamental variables in urban expansion and land use efficiency studies. For instance, Chen et al. [71] examined how changes in housing availability due to an increase in immigration affect mobility of labor, urban infrastructures, and financial affordability. The study showcases how a decent housing system that offers high-quality housing can help to shape urban development systems. Based on how second- and third-level variables interact, first-level variables could be improved, degraded, or remain unchanged.

## 5. Conclusions

Sustainable development logic has been widely applied to urban expansion issues since the Sustainable Development Goals were enacted. A city plan is based on adjusting land cover and dedicating city lands to urban uses, which means that assessments based on land cover transformations are necessary for adjusting the orientation of human interference and management decisions. Accommodating population growth, social organizing, and appropriately meeting the needs of citizens through improving service provision, in particular education and sport [72] and recreational facilities [73], are the minimum requirements for sustainable urbanism. Meanwhile, responding to population growth has always been among the main justifications for urban growth and expansion, which affect the balance of natural environments by transforming land cover. As a consequence, environmental carrying capacity is assumed to strike a balance between land use planning and ecosystem services. This kind of balanced urban development provides not only welfare and livability for residents, but also lowers urban living costs and protects natural resources. In this study, this was the dominant logic and approach used to improve the land use efficiency evaluation methodology suggested in the SDG 11.3.1 indicator.

Using demographic data from the Iran statistical center and Landsat satellite images classified according to land cover, SDG indicator 11.3.1 for Tehran was calculated over a 35-year period from 1986 to 2021 at 5-year intervals. During this period, Tehran was always expanding and its population was increasing. Urban land use allocation should be aimed at meeting urban needs, according to the main logic of this indicator. In order to assess the extent to which this goal was achieved during the study period, a comparison was made between this indicator and the trend of city services per capita and the number of neighborhoods with insufficient services. The comparison showed that Tehran's expansion did not improve services per capita and their accessibility and availability. Moreover, using the analysis of satellite images, LST and NDVI indexes were calculated at the beginning and end of the studied period. According to the results, the average LST has increased from 28 degrees centigrade in 1986 to 36 degrees centigrade in 2021, while the NDVI index has decreased from 0.228 to 0.204 over this period. These indexes show declining ecosystem services of green space and increased pressure on city resources.

In the final step, a content analysis of the literature on land use efficiency assessment was conducted and ten variables were extracted using the Delphi method. Then, each variable was examined for its effectiveness and affectability and placed in a three-level land use efficiency assessment framework developed using the DEMATEL and ISM methods. The third level includes the most effective and vital variables in the proposed framework. In other words, further land use planning or land use efficiency evaluations should prioritize variables located at this level. The variables at the second level rank second in importance and the variables at the first level are the most influential ones. According to these results, the most effective variables are transportation (A6), infill development (A7), and entrepreneurship (A10), which belong to the third level. In light of the fact that high levels of entrepreneurship, income, and standards of living can be potential drivers of land consumption, it seems that the degree of economic development has some bearing on land use. In other words, more financial prosperity can affect housing and infrastructure demand and be an incentive to increase housing supply, land consumption, and ultimately, urban expansion. The first-level variables, including land-surface temperature, air pollution, sewage and waste, water resources, and vegetation, are the most affectable variables.

These variables are usually at the core of urban planning policies and decision-making processes in Tehran and many other cities. However, urban officials in such cities should be informed that tackling high levels of LST and urban heat islands, air pollution, urban sewage and waste, and lack of resources and green areas are the main causes of unsustainable development. Instead, to appropriately address these challenges within the shortest possible time they have no choice but to focus on the main causes of these effects, i.e., the third- and second-level variables.

This paper contributes to the field of urban land use planning by improving the calculation model of the SDG's indicator 11.3.1 and also providing a framework for land use efficiency assessment based on ecosystem services. Because this framework is based on the most critical variables related to urban expansion, it can be used in other cities facing a fast-paced urbanization process accompanied by unplanned urban expansion. By using this framework, they can localize and improve their land allocation suitability rather than just using the ratio of land consumption rate to population growth for their evaluations. Moreover, a framework such as this can also be used by the United Nations to modify and improve the formula developed for indicator 11.3.1 of SGD 11. It is recommended to conduct further research to predict the possible and desirable direction of Tehran's future expansion. To do so, simulation tools and machine learning techniques can be employed in conjunction with the simultaneous analysis of satellite images and economic, social, and environmental factors influencing urban growth. By conducting such a study, urban managers and planners can guide the growth of the city and provide the required urban services while maintaining and improving ecosystem services.

**Author Contributions:** Conceptualization, S.T., S.A.A. and S.E.; methodology, S.T., S.A.A. and L.M.; software, S.T., S.A.A. and L.M.; validation, S.T., S.A.A., S.E., L.M. and A.S.; formal analysis, S.T., S.A.A. and S.E.; investigation, S.T., S.A.A., S.E., L.M. and A.S.; resources, S.T., S.A.A., S.E. and A.S.; data curation, S.T., S.A.A. and S.E.; writing—original draft preparation, S.T. and S.A.A.; writing—review and editing, S.T., S.A.A. and S.E.; visualization, S.T., S.A.A., S.E., L.M. and A.S.; supervision, S.T., S.A.A., S.E. and A.S.; project administration, S.T., S.A.A. and S.E. All authors have read and agreed to the published version of the manuscript.

**Funding:** This research received no external funding.

**Institutional Review Board Statement:** Not applicable.

**Informed Consent Statement:** Not applicable.

**Data Availability Statement:** Not applicable.

**Conflicts of Interest:** The authors declare no conflict of interest. The funders had no role in the design of the study; in the collection, analyses, or interpretation of data; in the writing of the manuscript; or in the decision to publish the results.

**Appendix A**

DEMATEL method data tables:

**Table A1.** Fuzzy direct correlation matrix.

|  | A1 | A2 | A3 | A4 | A5 | A6 | A7 | A8 | A9 | A10 |
|---|---|---|---|---|---|---|---|---|---|---|
| A1 | (0.000, 0.000, 0.000) | (0.467, 0.717, 0.917) | (0.367, 0.617, 0.833) | (0.583, 0.833, 0.950) | (0.583, 0.833, 0.967) | (0.333, 0.583, 0.817) | (0.283, 0.517, 0.767) | (0.267, 0.517, 0.767) | (0.400, 0.650, 0.850) | (0.283, 0.517, 0.750) |
| A2 | (0.483, 0.733, 0.900) | (0.000, 0.000, 0.000) | (0.450, 0.700, 0.883) | (0.450, 0.700, 0.867) | (0.517, 0.767, 0.933) | (0.333, 0.583, 0.800) | (0.333, 0.583, 0.817) | (0.383, 0.633, 0.867) | (0.417, 0.650, 0.883) | (0.333, 0.567, 0.783) |

**Table A1.** *Cont.*

|     | A1 | A2 | A3 | A4 | A5 | A6 | A7 | A8 | A9 | A10 |
|-----|----|----|----|----|----|----|----|----|----|-----|
| A3 | (0.267, 0.517, 0.750) | (0.450, 0.700, 0.900) | (0.000, 0.000, 0.000) | (0.533, 0.783, 0.933) | (0.550, 0.800, 0.950) | (0.233, 0.483, 0.700) | (0.317, 0.550, 0.767) | (0.283, 0.533, 0.783) | (0.333, 0.583, 0.817) | (0.233, 0.467, 0.717) |
| A4 | (0.500, 0.750, 0.933) | (0.317, 0.567, 0.800) | (0.400, 0.650, 0.867) | (0.000, 0.000, 0.000) | (0.600, 0.850, 0.983) | (0.200, 0.433, 0.667) | (0.500, 0.750, 0.967) | (0.433, 0.683, 0.883) | (0.600, 0.850, 0.983) | (0.383, 0.617, 0.817) |
| A5 | (0.600, 0.833, 0.950) | (0.600, 0.850, 0.983) | (0.433, 0.683, 0.900) | (0.567, 0.817, 0.967) | (0.000, 0.000, 0.000) | (0.300, 0.533, 0.767) | (0.350, 0.600, 0.817) | (0.433, 0.683, 0.883) | (0.383, 0.633, 0.850) | (0.283, 0.517, 0.750) |
| A6 | (0.583, 0.833, 0.967) | (0.667, 0.917, 1.000) | (0.333, 0.567, 0.767) | (0.267, 0.517, 0.750) | (0.400, 0.650, 0.867) | (0.000, 0.000, 0.000) | (0.467, 0.717, 0.917) | (0.483, 0.733, 0.917) | (0.517, 0.767, 0.950) | (0.467, 0.717, 0.933) |
| A7 | (0.417, 0.667, 0.867) | (0.433, 0.683, 0.867) | (0.433, 0.683, 0.900) | (0.450, 0.700, 0.900) | (0.417, 0.667, 0.883) | (0.467, 0.717, 0.900) | (0.000, 0.000, 0.000) | (0.517, 0.767, 0.933) | (0.567, 0.817, 0.983) | (0.467, 0.717, 0.900) |
| A8 | (0.350, 0.600, 0.850) | (0.383, 0.633, 0.883) | (0.283, 0.533, 0.783) | (0.383, 0.633, 0.867) | (0.333, 0.583, 0.817) | (0.417, 0.667, 0.917) | (0.450, 0.700, 0.933) | (0.000, 0.000, 0.000) | (0.450, 0.700, 0.917) | (0.383, 0.633, 0.867) |
| A9 | (0.500, 0.750, 0.967) | (0.500, 0.750, 0.933) | (0.583, 0.833, 0.967) | (0.567, 0.817, 0.950) | (0.533, 0.783, 0.950) | (0.567, 0.817, 0.967) | (0.517, 0.767, 0.950) | (0.517, 0.767, 0.950) | (0.000, 0.000, 0.000) | (0.450, 0.700, 0.900) |
| A10 | (0.417, 0.650, 0.867) | (0.383, 0.617, 0.850) | (0.400, 0.633, 0.833) | (0.367, 0.617, 0.833) | (0.250, 0.483, 0.717) | (0.483, 0.733, 0.967) | (0.433, 0.667, 0.900) | (0.467, 0.700, 0.900) | (0.517, 0.767, 0.967) | (0.000, 0.000, 0.000) |

**Table A2.** Complete correlation fuzzy matrix.

|     | A1 | A2 | A3 | A4 | A5 | A6 | A7 | A8 | A9 | A10 |
|-----|----|----|----|----|----|----|----|----|----|-----|
| A1 | (0.040, 0.181, 1.118) | (0.092, 0.261, 1.225) | (0.077, 0.237, 1.168) | (0.105, 0.273, 1.215) | (0.105, 0.274, 1.223) | (0.068, 0.222, 1.134) | (0.067, 0.225, 1.172) | (0.066, 0.229, 1.179) | (0.084, 0.253, 1.226) | (0.063, 0.213, 1.117) |
| A2 | (0.094, 0.263, 1.229) | (0.041, 0.187, 1.143) | (0.087, 0.248, 1.187) | (0.092, 0.263, 1.222) | (0.099, 0.270, 1.235) | (0.070, 0.226, 1.147) | (0.073, 0.234, 1.192) | (0.080, 0.244, 1.203) | (0.087, 0.256, 1.244) | (0.069, 0.222, 1.134) |
| A3 | (0.066, 0.228, 1.159) | (0.086, 0.248, 1.182) | (0.032, 0.159, 1.040) | (0.095, 0.257, 1.173) | (0.097, 0.259, 1.181) | (0.055, 0.202, 1.085) | (0.067, 0.218, 1.133) | (0.064, 0.220, 1.140) | (0.073, 0.235, 1.182) | (0.055, 0.199, 1.076) |
| A4 | (0.098, 0.272, 1.255) | (0.080, 0.256, 1.252) | (0.084, 0.251, 1.208) | (0.045, 0.195, 1.154) | (0.110, 0.286, 1.264) | (0.059, 0.218, 1.156) | (0.094, 0.258, 1.229) | (0.088, 0.256, 1.228) | (0.109, 0.284, 1.278) | (0.077, 0.233, 1.159) |
| A5 | (0.109, 0.280, 1.250) | (0.109, 0.283, 1.263) | (0.087, 0.253, 1.205) | (0.107, 0.282, 1.249) | (0.045, 0.195, 1.154) | (0.068, 0.226, 1.159) | (0.077, 0.242, 1.208) | (0.087, 0.255, 1.221) | (0.086, 0.261, 1.258) | (0.066, 0.222, 1.147) |
| A6 | (0.110, 0.288, 1.281) | (0.119, 0.299, 1.294) | (0.079, 0.249, 1.220) | (0.077, 0.259, 1.256) | (0.091, 0.273, 1.275) | (0.038, 0.176, 1.105) | (0.092, 0.262, 1.246) | (0.095, 0.268, 1.253) | (0.103, 0.283, 1.298) | (0.088, 0.250, 1.192) |
| A7 | (0.092, 0.272, 1.280) | (0.094, 0.276, 1.290) | (0.089, 0.261, 1.241) | (0.096, 0.278, 1.279) | (0.093, 0.275, 1.285) | (0.089, 0.253, 1.208) | (0.041, 0.185, 1.158) | (0.099, 0.272, 1.263) | (0.108, 0.289, 1.309) | (0.089, 0.250, 1.197) |
| A8 | (0.077, 0.244, 1.239) | (0.081, 0.249, 1.252) | (0.066, 0.226, 1.192) | (0.081, 0.250, 1.237) | (0.076, 0.245, 1.239) | (0.078, 0.230, 1.173) | (0.084, 0.241, 1.218) | (0.035, 0.170, 1.126) | (0.089, 0.256, 1.264) | (0.074, 0.224, 1.157) |

**Table A2.** *Cont.*

|  | A1 | A2 | A3 | A4 | A5 | A6 | A7 | A8 | A9 | A10 |
|---|---|---|---|---|---|---|---|---|---|---|
| A9 | (0.106, 0.296, 1.341) | (0.107, 0.299, 1.349) | (0.110, 0.290, 1.297) | (0.114, 0.306, 1.336) | (0.111, 0.302, 1.343) | (0.104, 0.277, 1.262) | (0.103, 0.282, 1.308) | (0.104, 0.287, 1.315) | (0.052, 0.217, 1.258) | (0.091, 0.262, 1.244) |
| A10 | (0.087, 0.254, 1.241) | (0.084, 0.253, 1.249) | (0.081, 0.241, 1.197) | (0.082, 0.253, 1.234) | (0.070, 0.240, 1.230) | (0.087, 0.241, 1.178) | (0.085, 0.243, 1.215) | (0.090, 0.250, 1.222) | (0.099, 0.268, 1.269) | (0.033, 0.159, 1.065) |

**Table A3.** De-fuzzification of the values of the complete correlation matrix.

|  | A1 | A2 | A3 | A4 | A5 | A6 | A7 | A8 | A9 | A10 |
|---|---|---|---|---|---|---|---|---|---|---|
| A1 | 0.341 | 0.42 | 0.392 | 0.429 | 0.431 | 0.374 | 0.383 | 0.387 | 0.413 | 0.365 |
| A2 | 0.423 | 0.349 | 0.404 | 0.421 | 0.429 | 0.379 | 0.394 | 0.403 | 0.419 | 0.374 |
| A3 | 0.384 | 0.404 | 0.311 | 0.41 | 0.413 | 0.351 | 0.372 | 0.375 | 0.393 | 0.374 |
| A4 | 0.434 | 0.42 | 0.408 | 0.358 | 0.446 | 0.374 | 0.418 | 0.416 | 0.445 | 0.386 |
| A5 | 0.439 | 0.444 | 0.41 | 0.44 | 0.357 | 0.381 | 0.402 | 0.414 | 0.424 | 0.375 |
| A6 | 0.45 | 0.46 | 0.408 | 0.422 | 0.436 | 0.333 | 0.423 | 0.429 | 0.447 | 0.404 |
| A7 | 0.436 | 0.441 | 0.421 | 0.441 | 0.439 | 0.41 | 0.35 | 0.434 | 0.453 | 0.405 |
| A8 | 0.409 | 0.416 | 0.387 | 0.413 | 0.41 | 0.387 | 0.404 | 0.333 | 0.421 | 0.379 |
| A9 | 0.463 | 0.466 | 0.451 | 0.47 | 0.469 | 0.435 | 0.447 | 0.452 | 0.389 | 0.42 |
| A10 | 0.417 | 0.418 | 0.399 | 0.415 | 0.404 | 0.396 | 0.404 | 0.411 | 0.431 | 0.314 |

**Table A4.** Threshold limit matrix.

|  | A1 | A2 | A3 | A4 | A5 | A6 | A7 | A8 | A9 | A10 |
|---|---|---|---|---|---|---|---|---|---|---|
| A1 | 0 | 0.42 | 0 | 0.429 | 0.431 | 0 | 0 | 0 | 0.413 | 0 |
| A2 | 0.423 | 0 | 0 | 0.421 | 0.429 | 0 | 0 | 0 | 0.419 | 0 |
| A3 | 0 | 0 | 0 | 0.41 | 0.413 | 0 | 0 | 0 | 0 | 0 |
| A4 | 0.434 | 0.42 | 0.408 | 0 | 0.446 | 0 | 0.418 | 0.416 | 0.445 | 0 |
| A5 | 0.439 | 0.444 | 0.41 | 0.44 | 0 | 0 | 0 | 0.414 | 0.424 | 0 |
| A6 | 0.45 | 0.46 | 0.408 | 0.422 | 0.436 | 0 | 0.423 | 0.429 | 0.447 | 0 |
| A7 | 0.436 | 0.441 | 0.421 | 0.441 | 0.439 | 0.41 | 0 | 0.434 | 0.453 | 0 |
| A8 | 0.409 | 0.416 | 0 | 0.413 | 0.41 | 0 | 0 | 0 | 0.421 | 0 |
| A9 | 0.463 | 0.466 | 0.451 | 0.47 | 0.469 | 0.435 | 0.447 | 0.452 | 0 | 0.42 |
| A10 | 0.417 | 0.418 | 0 | 0.415 | 0 | 0 | 0 | 0.411 | 0.431 | 0 |

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
