# Peer review of "Evaluation of Land Use Efficiency in Tehran’s Expansion between 1986 and 2021: Developing an Assessment Framework Using DEMATEL and Interpretive Structural Modeling Methods"

_sustainability, doi:10.3390/su15043824_

Round 1

Reviewer 1 Report

These patterns in article are significant in arranging population attraction, territory planning, and urban management due to the mutual relationship with providing  urban services per capita for accessibility, citizen welfare and livability from one side  and consuming natural resources and using ecosystem services from the other side. Urban expansion has always been considered one of the most critical  challenges in land use planning, since urban sprawl and low-density development, as  well as geographical divisions of essential land uses  have resulted in an increase in public and private costs. That's why the role of urban management is evident in determining frameworks of population attraction and distribution patterns in urban areas,  along with planning land use and determining urban services per capita and accessibility to have the maximum efficiency while spending the minimum cost. SDG 11 emphasizes the sustainable development of urban areas. This goal has considered the challenges of providing affordable housing, access to  sustainable transportation in the cities, access to open green spaces, spatial justice, and  eradicating urban poverty and land use efficiency. Generally, land use efficiency studies can be divided into four categories: calculations, location and time characteristics, contributing variables, and improvement methods.T his study takes into account the accuracy and localization of indicator  11.3.1 in order to optimize the assessment of land use and consumption efficiency in  Tehran, as well as manage land use.

The article covers a wide period of study, even 35 years, and evaluates the territorial development of the city of Tehran. Regarding sustainability, it is necessary to emphasize that ten indicators were selected from the analysis of literature sources in order to assess the significance of their impact on the development of the city. Data from the Delfi expert survey (14 experts) were used to determine the significance of individual indicators, which are divided into three groups. The calculations and illustrations fully reflect the conducted research, but I missed the description of the 14 selected experts, the justification of the compatibility of their opinions.

Using the latest data from Tehran municipality, the Statistical Center of Iran, and  Landsat satellite photos, this paper aims to improve land use efficiency evaluation and  land consumption frameworks.

How can the dependencies identified in the article be applied to the development assessment of other cities, or can these results be applied only to a limited extent.

The structure of the article should be better balanced, too much emphasis is placed on describing the situation.

Author Response

Response to Reviewer 1 Comments

Thank you very much for your kind and valuable comments. The paper has been totally revised and upgraded regarding research design, questions, hypotheses, methods, conclusions, theoretical background and empirical research.

Point 1: The calculations and illustrations fully reflect the conducted research, but I missed the description of the 14 selected experts, the justification of the compatibility of their opinions.

Response 1: It has been added to section “3.2.2. Preparation and completion of a questionnaire using the Delphi method” in the revised manuscript.

Point 2: How can the dependencies identified in the article be applied to the development assessment of other cities, or can these results be applied only to a limited extent.

Response 2: More explanation on this issue has been added to the last paragraph of the Conclusion section in the revised manuscript.

Point 3: The structure of the article should be better balanced, too much emphasis is placed on describing the situation.

Response 3: The structure of the paper has been revised.

Reviewer 2 Report

Referee report on "Evaluation of land use efficiency in Teheran’s Expansion between 1986 and 2012: development an assessment framework using DEMANTL and interpretive structural modeling methods"

The paper in front of us provides an interesting and informative contribution. It is written well and easy to follow. Overall, I’m sympathetic regarding the paper. My comments are relatively minor and mostly relate to the exposition.

Comments:

1. Your period starts from 1986, but you never explain the choice of your starting date. Please provide an explanation somewhere.

2. When you write that “the urban centers may attract a larger population and expand rapidly” (lines 66 – 67), you could add the following statement: “which is likely to increase urban congestion diseconomies (Azarnert, 2019; Yakubenko, 2020).”

“Congestion diseconomies” is the professional term that is usually used in urban economics to describe all potential ills from overconcentration.

3. Strengthen your description of Teheran in Chapter 2 (Study Area). Add more socio-economic data on Teheran. Provide some figures on Teheran’s contribution to the economy as a whole and add some info about industrial production, wages and standard of living on the general in this city relative to the rest of the country.

4. When you present the seven main urban services in Tehran, provide some more details on these services.

5. Also describe in a bit more detail the socioeconomic category that you refer to in Table 4 and further on.

5a. In particular, add one sentence or two on “population attraction”.

6. In Chapter 4.1 you mention that 21% of population growth in Teheran is due to immigration. Please provide a short paragraph to describe the Iranian official policies with respect to immigration. I never heard about any particular restrictions, as for example, the Chinese Hukou system, but was the official policy more pro- or anti-migration? What is the origin of these immigrants, their levels of education and/ or socio-economic composition etc…?

6a. You’ve also mentioned the so-called Construction Period. Add a couple of sentences to provide a bit more details.

7. Re-check the line that describes the trend in “water” in Figure 4. As we learn from Table 7, there was a serious improvement over the sub-period of 2011 – 2016, but it is not reflected in the Figure.

8. In line 374 you inform us that Teheran has never experienced urban shrinking. Could you let us know were there any small- medium-size cities in Iran that experienced urban shrinking? For example in China, this phenomenon was observed in a number of such cities that were the source of outmigration to rapidly growing megacities.

9. In Table 9 you speak about the upward trend in the temperature. What is the scale that you use there? Maybe you could also refer to more standard Celsius degrees?

10. In line 529, Chen is not the only author of the mentioned study. Refer to all of them as Chen et al.

11. When you refer to services supply in line 549, you can add “in particular, education and sport (Azarnert, 2020) and recreational facilities (Dahmann et al., 2010)”

12. There are a few linguistic irregularities that should be fixed.

Overall, it is an interesting paper and I expect that the readers will also like it.

 References

Dahmann, N., Wolch, J., Joassart-Marcelli, P., Reynolds, K., and Jerrett, M. (2010) The active city? Disparities in the provision of urban recreation resources, Health and Place, 16, 431–445

Azarnert, L.V. (2019) Migration, congestion and growth. Macroeconomic Dynamics 23(8), 3035–3064

Azarnert, L.V. (2020) Health capital provision and human capital accumulation. Oxford Economic Papers, Vol. 72(3), 633–650

Yakubenko, S. (2020) Giants and midgets: The effect of public goods provision on urban population concentration. Cities, 107, 102872

Author Response

Response to Reviewer 2 Comments

Thank you very much for your kind and valuable comments. The paper has been totally revised and upgraded regarding research design, questions, hypotheses, methods, conclusions, theoretical background and empirical research.

Point 1: Your period starts from 1986, but you never explain the choice of your starting date. Please provide an explanation somewhere.

Response 1: More explanation has been added to section “3.1. Tehran’s land use efficiency assessment according to SDG11.3.1 Indicator.”

Point 2: When you write that “the urban centers may attract a larger population and expand rapidly” (lines 66 – 67), you could add the following statement: “which is likely to increase urban congestion diseconomies (Azarnert, 2019; Yakubenko, 2020).”

“Congestion diseconomies” is the professional term that is usually used in urban economics to describe all potential ills from overconcentration.

Response 2: The recommended sentence and references have been added to the revised manuscript.

Point 3: Strengthen your description of Teheran in Chapter 2 (Study Area). Add more socio-economic data on Teheran. Provide some figures on Teheran’s contribution to the economy as a whole and add some info about industrial production, wages and standard of living on the general in this city relative to the rest of the country.

Response 3: The Study Area section has been updated with additional data on Tehran's socioeconomic characteristics and some reasons for its population growth.

Point 4: When you present the seven main urban services in Tehran, provide some more details on these services.

Response 4: A more detailed explanation has been added to the paragraph before Table 2.

Point 5: Also describe in a bit more detail the socioeconomic category that you refer to in Table 4 and further on. In particular, add one sentence or two on “population attraction”.

Response 5:

A more detailed explanation has been added to the paragraph before Table 4.

Point 6: In Chapter 4.1 you mention that 21% of population growth in Teheran is due to immigration. Please provide a short paragraph to describe the Iranian official policies with respect to immigration. I never heard about any particular restrictions, as for example, the Chinese Hukou system, but was the official policy more pro- or anti-migration? What is the origin of these immigrants, their levels of education and/ or socio-economic composition etc…?

You’ve also mentioned the so-called Construction Period. Add a couple of sentences to provide a bit more details.

Response 6: Because this kind of information can change the direction of a paper, the authors decided to delete it from the revised manuscript. About the Construction Period, a sentence has been added to section "4.1.1. Land cover classification using Landsat satellite images"

Point 7: Re-check the line that describes the trend in “water” in Figure 4. As we learn from Table 7, there was a serious improvement over the sub-period of 2011 – 2016, but it is not reflected in the Figure.

Response 7: We have replaced the figure with the correct one.

Point 8: In line 374 you inform us that Teheran has never experienced urban shrinking. Could you let us know were there any small- medium-size cities in Iran that experienced urban shrinking? For example in China, this phenomenon was observed in a number of such cities that were the source of outmigration to rapidly growing megacities.

Response 8: In Iran, as in other developing countries, the existence of shrinking cities has been observed. This phenomenon is more common in cities of disadvantaged provinces that suffer from net outmigration and the devastating effects of the Iran-Iraq war, climate change, lack of water resources, the loss of the possibility of agriculture and socio-economic inequalities accompanied by the attractiveness of some major cities such as Tehran, Esfahan, Mashhad and Shiraz. Among these provinces are Khuzestan and Sistan and Baluchestan.

Point 9: In Table 9 you speak about the upward trend in the temperature. What is the scale that you use there? Maybe you could also refer to more standard Celsius degrees?

Response 9: Celsius signs have been added to Table 10 in the revised manuscript.

Point 10: In line 529, Chen is not the only author of the mentioned study. Refer to all of them as Chen et al.

Response 10: Corrected

Point 11: When you refer to services supply in line 549, you can add “in particular, education and sport (Azarnert, 2020) and recreational facilities (Dahmann et al., 2010)”

Response 11: The recommended sentence and references have been added to the revised manuscript.

Point 12: There are a few linguistic irregularities that should be fixed.

Response 12: We have improved the writing style and corrected linguistic and editing errors in the manuscript.

Reviewer 3 Report

The analysis was documented and carried out in a clear and systematic way, but there are editorial errors, e.g. repeated whole sentences or fragments from elsewhere in the argument. The text should be edited in this respect.

Author Response

Response to Reviewer 3 Comments

Thank you very much for your kind and valuable comments. The paper has been totally revised and upgraded regarding research design, questions, hypotheses, methods, conclusions, theoretical background and empirical research.

Point 1: The analysis was documented and carried out in a clear and systematic way, but there are editorial errors, e.g. repeated whole sentences or fragments from elsewhere in the argument. The text should be edited in this respect.

Response 1: We have improved the writing style and corrected linguistic and editing errors in the manuscript.